# Imaging of Pulmonary Sarcoidosis—A Review

**DOI:** 10.3390/jcm13030822

**Published:** 2024-01-31

**Authors:** Georgina L. Bailey, Athol U. Wells, Sujal R. Desai

**Affiliations:** 1Department of Radiology, Royal Brompton Hospital, London SW3 6NP, UKs.desai@rbht.nhs.uk (S.R.D.); 2The Interstitial Lung Disease Unit, Royal Brompton Hospital, London SW3 6NP, UK; 3The National Heart & Lung Institute, Imperial College London, London W12 7RQ, UK; 4The Margaret Turner-Warwick Centre for Fibrosing Lung Diseases, Imperial College London, London W12 7RQ, UK

**Keywords:** thoracic, pulmonary, sarcoidosis, imaging

## Abstract

Sarcoidosis is the classic multisystem granulomatous disease. First reported as a disorder of the skin, it is now clear that, in the overwhelming majority of patients with sarcoidosis, the lungs will bear the brunt of the disease. This review explores some of the key concepts in the imaging of pulmonary sarcoidosis: the wide array of typical (and some of the less common) findings on high-resolution computed tomography (HRCT) are reviewed and, with this, the concept of morphologic/HRCT phenotypes is discussed. The pathophysiologic insights provided by HRCT through studies where morphologic abnormalities and pulmonary function tests are compared are evaluated. Finally, this review outlines the important contribution of HRCT to disease monitoring and prognostication.

## 1. Introduction

Sarcoidosis, the archetypal granulomatous disease, was first reported in the 19th century by the physician Jonathan Hutchinson [1]. For a while thereafter, sarcoidosis was considered a disorder of the skin. However, the multisystem nature of sarcoidosis was soon realised, and it also became clear that the lungs bear the brunt in most patients [2,3,4]. The cardinal diagnostic finding on histopathologic examination is the non-necrotising or non-caseating epithelioid cell granuloma [5,6]. Yet, despite the commonality of pathologic features, it is also widely known that patterns of functional impairment, responses to treatment and prognosis can vary considerably from patient to patient [7,8,9,10]. Indeed, because of this, it has been posited that sarcoidosis might simply be a convenient ‘umbrella’ capturing what, in essence, are multiple different granulomatous diseases.

Imaging tests play a role not only in diagnosis but also in management and follow-up. In the review that follows, we consider the common and some atypical patterns of lung involvement in sarcoidosis. We also discuss the potential place of computed tomography (CT) in ‘staging’, quantification of disease extent (leading to discussions on prognostication) and clinical monitoring in sarcoidosis.

## 2. Imaging in Sarcoidosis—General Principles

Kuznitsky and Bittorf first reported the plain chest radiographic (CXR) abnormalities in sarcoidosis early in the first half of the 20th century [11]. Attempts to better characterise CXR appearances, first proposed by Wurm [12], were later modified by Scadding [13]. In Scadding’s modification, CXR abnormalities are stratified based on the presence or absence of intra-thoracic nodal enlargement and parenchymal disease. Despite the simplicity, the clinical utility of this CXR staging system has been questioned: linkages with functional tests [14,15] and patient-reported disease severity [16] are, at best, weak. The imperfect interobserver agreement further limits the value of CXR staging [17]. Finally, it is worth emphasising that even the use of the term staging is misleading; in contrast to malignant disease, there is no predictable, stepwise progression from ‘lower’ to ‘higher’ stages in sarcoidosis [18].

### 2.1. Imaging in Sarcoidosis: Plain CXR vs. High-Resolution Computed Tomography

The last two decades have seen significant advances in imaging technologies. Yet, the plain CXR, computed tomography (CT) and, specifically, high-resolution CT (HRCT), remain the mainstays of imaging tests for interstitial lung disease. Plain CXR has the benefits of relative technical simplicity, high spatial resolution, reasonably low cost and a limited radiation burden. Against this, contrast resolution in CXR is lower than in CT, and anatomical superimposition on CXR images also hampers diagnostic interpretation.

The advent of HRCT was a major step forward in the diagnosis of diffuse interstitial lung diseases (DILDs): compared with standard CT images, spatial resolution and image quality, in general, were enhanced by reducing section thickness [19,20,21] and the use of a dedicated high-spatial-frequency (‘bone’) reconstruction algorithm [22]. The diagnostic potential of HRCT was realised in the pivotal study by Mathieson and co-workers in which three experienced, blinded observers independently reviewed CXRs and HRCT studies in 118 patients with DILDs [23]. The key findings were not only that observers were more than twice as confident in formulating a diagnosis with HRCT (23% versus 49%) but also that, when confident, the HRCT diagnosis was almost always correct. The advent of spiral volumetric and, subsequently, multidetector computed tomography scanning has facilitated the rapid (single breath-hold) acquisition of volumetric thin-section datasets yielding further improvements in image quality [24,25]. Volumetric—as opposed to interspaced—thin-section CT of the lungs is now the norm in most imaging departments. The reader should note that for the purposes of the current review, the abbreviation CT will be used to refer to volumetric HRCT acquisitions.

### 2.2. Imaging in Pulmonary Sarcoidosis: Other Imaging Modalities

Plain CXR and CT are almost always the first imaging tests requested in patients with suspected or established lung disease. In specific clinical scenarios, other imaging tests are brought to bear. Positron emission tomography (PET) using a radioactive tracer (most commonly radio-labelled fluorodeoxyglucose [^18^FDG]), is coupled with CT to pinpoint the foci of metabolically ‘active’ disease. Accordingly, in pulmonary sarcoidosis, ^18^FDG-PET/CT may be used to assess the presence and extent of active inflammation [26]. Indeed, diffusely increased PET avidity in lung parenchyma has been correlated with a significant decrease in diffusion capacity for carbon monoxide in sarcoidosis (Dlco) [27]. Away from the lungs, PET/CT has a more established role in the detection and monitoring of cardiac sarcoidosis, with a reported sensitivity and specificity of 89% and 78%, respectively [28,29]. PET/CT also has a potential role in detecting occult extra-thoracic disease in sarcoidosis [30,31,32].

Assessment of pulmonary disease with magnetic resonance imaging (MRI) is limited by several factors, including poor signal-to-noise ratio, significant susceptibility artefact at the interfaces between air and soft tissue, and respiratory and cardiac-related motion artefacts during long scanning times [33]. Despite technical developments in MRI, such as ultrashort echo times and parallel acquisition methods [34], the spatial resolution does not allow distinction between finer morphological features, for instance, differentiating reticulation from honeycombing [35]. As with PET/CT, MRI is more often utilised in the detection of cardiac sarcoidosis, with 95% and 85% sensitivity and specificity, respectively [36], and it is also sensitive, but not particularly specific, for neurosarcoidosis [37].

## 3. CT Detection and Diagnosis of Sarcoidosis

### 3.1. Intra-Thoracic Nodal Enlargement

Enlargement of mediastinal and hilar lymph nodes is a hallmark of sarcoidosis, reported on CT in up to 84–97% of cases [38,39,40,41], and most commonly involving stations 4R, 7, 11L and 11R [40]; the classical Garland’s triad of bilateral hilar and right paratracheal nodal enlargement will be known to most readers [42]. Not surprisingly, the distribution and extent of nodal enlargement are best evaluated on CT [43]. On the whole, symmetrical hilar nodal enlargement most often points to a diagnosis of sarcoidosis and away from lymphoma, other malignancies and tuberculosis (TB); in TB, calcification is more often unilateral and along predictable lymphatic drainage pathways [44]. Necrosis of lymph nodes is recognised in sarcoidosis but should prompt a search for an alternative aetiology, such as TB [39]. Nodal calcification, present in 44–53% of patients, also tends to be bilateral and may have a focal pattern (as opposed to complete, asymmetrical nodal calcification which is more commonly observed in TB) [41,44]; so-called ‘egg-shell’ calcification is also reported [45]. An interesting variant is seen in some patients wherein the calcification has a more ill-defined or ‘icing sugar’ quality [44] (Figure 1).

Precise localisation of intra-thoracic lymph nodes on CT may facilitate the planning of endobronchial ultrasound-guided biopsy, a minimally invasive technique that can provide a more definitive diagnosis in ambiguous clinico-radiological presentations [46].

### 3.2. Nodules

Lung nodules on CT are the most widely recognised and common CT manifestations of sarcoidosis [7,47,48]. In a small series of 45 patients with suspected or known sarcoidosis, nodules were present in 80% [7]. In a larger study of 95 patients by Remy-Jardin et al., a nodular pattern was present in 93% [49]. In the classical case, bronchocentric micronodules (measuring 1–3 mm in diameter), are seen in the mid and upper zones. Lung nodules, corresponding to aggregates of microscopic epithelioid granulomata [50], may be diffusely distributed throughout the lungs or, less frequently, localised to one or several focal areas. The predilection for the axial interstitium (i.e., surrounding bronchovascular bundles), accounts for the readily recognisable thickened, irregular perihilar and peribronchovascular appearance [48] (Figure 2). Irregular or nodular thickening of interlobular septa—mimicking lymphangitis carcinomatosa—is recognised but is rarely a dominant feature [51,52,53].

Nodules along the subpleural surfaces (including the fissures) give rise to a characteristic beaded appearance [54]. Less often, centrilobular or branching nodularity is seen but this is usually in conjunction with a dominant pattern of bronchovascular nodules [55,56]. Extensive nodularity in a random distribution is present in some patients and, for obvious reasons, the distinction from disseminated TB or malignancy then not only becomes important but also difficult, often mandating histopathologic/microbiologic confirmation [57].

### 3.3. Masses and Consolidation

On occasion, granulomata coalesce to form larger nodules or masses, sometimes manifesting as a pattern of consolidation [48,49]. Nodules measuring over 1 cm in maximum diameter have been reported in 15–53% of patients; these nodules tend to have irregular margins and predominate in the mid and upper zones [50,58,59,60]. Despite the occasional presence of air bronchograms (giving the impression of airspace involvement), the large nodules and appearance of ‘consolidation’ are a consequence of conglomerated granulomata and/or interstitial thickening as opposed to filling of the alveoli [50,61].

Clustering of micronodules around a larger central nodule gives rise to the so-called ‘galaxy sign’ [62], (Figure 3) reportedly seen in nearly one-third of patients [63], but not pathognomonic for sarcoidosis [64].

### 3.4. Ground-Glass Opacification

The reported prevalence of ground-glass opacification on CT in sarcoidosis is highly variable, ranging from 16 to 42%, with most instances of this pattern occurring in conjunction with other more common CT features [7,47,49]. Indeed, in the recent multinational Delphi study of recognisable CT phenotypes in sarcoidosis, there was no consensus, among a large body of experts, as to the existence of a predominant pattern of ground-glass opacification [65]. When present in sarcoidosis, ground-glass opacities most commonly reflect multiple microscopic granulomata [50].

### 3.5. Airway Disease

Airway involvement in sarcoidosis is more prevalent on CT than generally appreciated; the putative pathogenetic mechanisms of airway disease include inflammation, constriction related to surrounding fibrosis and, in some cases, extrinsic compression. Non-specific and mild bronchial wall thickening may be seen in nearly two-thirds of patients and correlates with the presence of bronchial granulomata, erythema and oedema on endoscopy, evolving to fibrotic bronchial stenosis in up to 14% [66,67]. In addition, the formation of granulomata along the axial interstitium of the bronchovascular bundles may lead to extrinsic airway narrowing. In fibrotic sarcoidosis, the airways may be distorted and abnormally dilated by surrounding retractile fibrosis (i.e., traction bronchiectasis) (Figure 4a,b).

Involvement of the small airways is a surprisingly common finding on CT in sarcoidosis: subtle mosaicism—reflecting obliterative bronchiolitis—is often visible and enhanced on images obtained at end-expiration [68]. Limited involvement of less than 25% of the lung is likely to be clinically insignificant, but air trapping is reported on expiratory phase CT in the majority of patients [7,69,70,71] (Figure 5).

### 3.6. Pulmonary Fibrosis

Pulmonary fibrosis develops in 20–30% of patients [13,72]. The typical CT manifestations include coarse linear opacities, bronchocentric reticulation causing volume loss in the upper lobes and classical posterior retraction of the central bronchovascular structures [48,58] (Figure 6). Encasement of the bronchovascular bundles with conglomerate fibrosis masses may occur, with bronchial distortion and traction bronchiectasis/bronchiolectasis [10,47,73]. Honeycombing is seen in a significant minority and, in contrast to idiopathic pulmonary fibrosis (IPF), has a predilection for the mid-to-upper zones [10,74]. That said, in some patients, sarcoidosis does appear to masquerade as IPF on CT with basal predominant reticulation, ground-glass opacification and interlobular septal thickening [75]. In a recent study by Collins et al., 25 patients with combined sarcoidosis and IPF were reviewed [76]. Interestingly, the diagnosis of sarcoidosis was made, on average, a decade earlier than IPF; in 68%, sarcoidosis had been diagnosed on histopathologic examination at the time of IPF diagnosis. More importantly, survival in patients with combined disease was comparable to patients with classical IPF. Reports such as this raise the question of whether patients with combined disease represent a novel sarcoid phenotype or simply reflect a chance association (i.e., with IPF developing in patients with established sarcoidosis) [77].

## 4. Uncommon CT Manifestations and Complications in Pulmonary Sarcoidosis

### 4.1. Cavitation

Cavitation in sarcoidosis is uncommon and seen in ~10% of patients with advanced disease [78]. Primary cavitary sarcoidosis is estimated to affect around 2% and, again, tends to occur in patients with severe, ‘active’ sarcoidosis [79]. Superimposed infection (particularly with fungi or mycobacterial species) should always be considered in this context.

### 4.2. Fungal Colonisation

Fungal colonisation, most commonly with *Aspergillus* species, complicates between 3 and 12% of sarcoidosis cases with fibrocavitary (or fibrobullous) disease [80]. The radiologic manifestation might be in the form of a simple aspergilloma within a densely fibrotic lung, within a pre-existing bulla or grossly ectatic airway [81,82] (Figure 7). Serological and biochemical markers may be of value in diagnosis [83,84]. In a minority of patients, untreated fungal colonisation will lead, over time, to chronic and extensive fibrotic destruction [81].

### 4.3. Pleural Disease

Although generally considered rare, Szwarcberg et al. found that in a study of 61 patients with sarcoidosis, 41% had evidence of pleural involvement, predominantly in the form of pleural thickening, and that this was associated with restrictive pulmonary dysfunction [85]. However, it is possible that inward retraction of the pleura and extrathoracic soft tissue in the context of fibrotic pulmonary sarcoidosis might mimic pleural thickening, and interstitial fibrosis also accounts for restrictive functional abnormality in some cases. Pleural effusions are observed in under 10% of sarcoidosis patients [85,86]; reports of pneumothorax are limited to case reports in the literature and are mostly accounted for as a complication of bullous disease [87,88,89,90].

### 4.4. Pulmonary Hypertension

Pulmonary hypertension (PH), defined as mean pulmonary pressures above 20 mmHg [91], affects between 5.7 and 12% of sarcoidosis patients and is associated with significantly reduced pulmonary function [92,93,94]. While predominantly affecting those with CXR Stage IV disease, sarcoidosis-associated PH (SAPH) is not limited to patients with fibrosis [95]. The pathophysiology of SAPH is multifactorial, including granulomatous involvement of the vessel walls, vasoconstriction due to fibrosis and venous occlusion secondary to lymphatic granulomas [96,97]. PH may also follow left heart disease in patients with cardiac sarcoidosis.

Mean pulmonary artery diameter measurement (MPAD) of more than 29 mm or a ratio of the diameter relative to the ascending aorta greater than 1 is suggestive of raised pulmonary pressures (greater than 25 mmHg) and should be considered in decisions concerning the need for formal assessment for PH [98]. Another feature suggestive of PH on CT is a segmental artery-to-bronchus ratio greater than 1 in three of four lobes [99]. In a small study by Nunes et al., septal lines were more frequently seen in patients with fibrotic sarcoidosis and PH than in those with fibrotic sarcoidosis without PH [100]. While CT may prompt further workup, the absence of the described features does not exclude PH in patients with sarcoidosis.

### 4.5. Halo/Reversed-Halo Sign

The ‘halo sign’ on CT comprising a central nodule (or consolidation) with surrounding ground glass opacification—also found in other pathologies (Including angioinvasive aspergillosis and hypervascular metastases [101,102,103])—is an infrequent manifestation in sarcoidosis, corresponding to aggregates of macrophages in the alveolar spaces surrounding sarcoid granulomata [104]. A variant of this sign, the ‘reversed halo’ or ‘atoll’ sign (once touted as a highly specific sign for organising pneumonia [105]), is also recognised in sarcoidosis, albeit rarely [106,107].

## 5. Disease Monitoring in Pulmonary Sarcoidosis

In any disease, monitoring seeks to identify patients with severe and/or progressive disease which is almost inevitably associated with poorer outcomes [108,109,110]. With regard to sarcoidosis—and, for that matter, any other interstitial lung disease (ILD)—it is also worth stressing that monitoring disease behaviour where previously only a provisional or ‘working’ diagnosis was possible, might confirm the initial suspicion or, at least, suggest diagnostic alternatives. As highlighted previously, one of the bigger challenges in sarcoidosis is the heterogeneous nature of sarcoidosis: in many patients, complete resolution occurs (or, at least, there is stabilization without treatment) whereas others face inexorable deterioration culminating in end-stage fibrosis [111,112]. Indeed, the notion of sarcoidosis as a ‘benign’ disorder is questionable, particularly given a recent large registry review [113]. Hambly and co-authors showed that just under one-third of 92 patients with sarcoidosis fulfilled the criteria for progression (as per the INBUILD trial parameters [114]). That said, in contrast with IPF, fibrotic hypersensitivity pneumonitis, ILDs related to connective tissue disease and even unclassifiable ILDs, the intrinsic likelihood of progression in sarcoidosis is lower. In sarcoidosis, this has implications not only for monitoring but also for the setting of satisfactory ‘thresholds’ by which progression is to be judged.

For most pulmonologists, establishing progression will be a three-pronged exercise: firstly, a symptomatic assessment, second, evaluation of serial changes in pulmonary function tests (PFTs) and, finally, review of imaging tests (principally CXR and CT). In this respect, it is worth stressing that while each might provide a clue, none is sufficiently sensitive or specific in isolation. Another key challenge for the pulmonologist is determining what constitutes significant change. A detailed critique of the advantages and limitations of clinical assessment and PFTs is not the focus of the present article. Suffice it to say that determining progression on the basis of patient-reported symptoms is not straightforward. For instance, worsening breathlessness, while being indicative of progression in some might, equally, be the harbinger of pulmonary hypertension or a consequence of infection associated with treatment. In contrast to symptomatic assessment, PFTs have the benefit of greater objectivity. Yet, here too, there are important considerations: for instance, minor serial changes in forced vital capacity (FVC), of ≤10%, in the absence of a decline in Dlco should be interpreted with caution. Another consideration is that the estimation of Dlco, an important physiologic marker of interstitial lung disease, is not consistent across laboratories, making the evaluation of serial change based on Dlco measurement more difficult.

Plain CXR and CT are the cornerstones of imaging assessment in sarcoidosis. The limitations of CXR have been discussed briefly above and the diagnostic advantage of CT is clear. Against this, it is worth emphasizing that the detection of a real change (for instance, in the patterns or extent of disease) on CXR is still clinically meaningful, especially so where serial changes in symptoms or function are equivocal. Admittedly, the exact place or utility of CT in monitoring disease has not been defined. Suffice it to say that any programme of monitoring sarcoidosis should probably also include a ‘baseline’ CT against which change might be judged even though, as it stands, no national or international guidelines recommend CT for this purpose. The latter situation may change following the publication recently of the Delphi-based position statement showing high-level agreement among experts on the need for baseline CT in patients with sarcoidosis and evidence of interstitial lung disease [65].

CT monitoring in sarcoidosis serves a number of purposes (Table 1). In some patients, the main issue will be to assess reversibility: in ‘classical’ nodular sarcoidosis, for instance, significant or even complete resolution might be expected. By contrast, with predominant upper zone bronchocentric fibrosis and volume loss, the prospects for improvement are likely to be lower. Other indications for requesting serial CT will be to assess the response to treatment and to identify those who progress despite management. With regard to the latter, the evaluation of progression on CT can be difficult and this is compounded by inter-/intra-observer variation and observer experience, to say nothing of the technical challenges of CT interpretation (e.g., variation between CT scanners, scan-to-scan differences in inspiratory effort, etc). Deciding what constitutes a significant change in CT also warrants brief discussion—minor differences in the overall CT extent of the abnormal lung are best disregarded, particularly in the absence of major symptomatic and/or functional decline. Another point to remember is that progression should not solely be defined by an increase in extent; a change in the pattern(s) of disease—for example, an increase in the severity of traction bronchiectasis over time (for the same overall extent of abnormality)—can also indicate that disease has progressed. 

## 6. CT Phenotypes in Sarcoidosis

There are few (if any) disorders of the lung with such a plethora of possible imaging manifestations. Added to this and given the considerable variability in functional parameters, natural history, treatment response or outcomes, it is tempting to speculate that the diagnostic label ‘sarcoidosis’ might simply refer to a multiplicity of entirely different diseases. With this background, a recent multinational study sought consensus from sarcoidosis experts on the existence of distinct morphological CT subtypes or ‘phenotypes’ of pulmonary sarcoidosis [65]. A total of 146 expert radiologists and pulmonologists from 28 countries took part in a Delphi study. Over two rounds—with ‘consensus’ defined as ≥70% agreement among observers—the study investigators achieved agreement on seven CT phenotypes comprising combinations of CT signs and patterns in sarcoidosis, broadly divided into ‘non-fibrotic’ and ‘likely to be fibrotic’ subtypes (Table 2). Further work in the field is certainly required to define the prevalence of different phenotypes (including those for which no consensus was reached), observer agreement for their recognition of CT and the physiological/prognostic impact, if any, of CT subtypes. However, studies of the type listed above might pave the way for a ‘new’ classification of sarcoidosis based on CT morphology which, in contrast with histopathologic features, may link more closely with observed physiologic and/or prognostic differences in sarcoidosis.

## 7. Disease Quantification and Prognostication in Sarcoidosis

### 7.1. Morphological–Functional Relationships in Sarcoidosis

In pulmonary sarcoidosis, pulmonary function tests (PFTs) may be entirely normal, but airflow obstructive, restrictive and mixed defects are widespread [8]. Not surprisingly, severe restrictive ventilatory defects are usually associated with extensive fibrosis [110,115,116]. However, an obstructive defect, which is not typically associated with fibrotic ILDs other than sarcoidosis, is also relatively common, even in patients with advanced fibrosis [15]. Diffusion capacity (Dlco) is reduced in as many as two-thirds of patients with sarcoidosis [117], variably reflecting interstitial disease and pulmonary vasculopathy [118].

The ability to characterize and quantify specific lung abnormalities on CT and relate these to functional indices or outcomes has provided unique pathophysiologic insights into many DILDs [119,120,121,122,123,124,125,126,127]. Similar structure–function studies have been undertaken in sarcoidosis. For instance, lung nodules in pulmonary sarcoidosis, for the most part, appear to be functionally ‘silent’ [7,49,128,129]. There are more intriguing linkages between a CT reticular pattern and functional tests in sarcoidosis: in the study by Hansell et al., reticulation was the dominant independent determinant of functional impairment, especially airflow obstruction [7]. Moreover, an unexpected finding was that the extent of reticulation was associated with indices of obstruction—more often than not, a CT reticular pattern implies lung fibrosis which would cause functional restriction. It should be stated that, in this same study, the extent of decreased attenuation (as part of a CT mosaic pattern) on expiratory imaging also correlated with obstructive impairment but the relationship was less strong than for reticulation [7].

In many patients with pulmonary sarcoidosis, a combination of CT patterns and signs co-exist. For instance, Abehsera et al. identified three patterns of fibrotic sarcoidosis based on the predominant lesions with very good interobserver agreement [10]. Pulmonary restriction with a low diffusion capacity was mostly associated with the honeycomb pattern, whereas obstructive indices were more often linked to bronchial distortion. Those with a linear pattern generally had less severe functional impairment, except in cases of ‘distorted septal reticulation’, which correlated with pulmonary hypertension, perhaps as a consequence of venous occlusion because of septal fibrosis [10].

### 7.2. Reversible, Irreversible and Progressive Disease in Sarcoidosis

Of the variety of CT patterns reported in sarcoidosis, nodular infiltrates are most likely to improve or resolve at follow-up [41]. Additionally, peribronchovascular thickening, consolidation and ground-glass opacification also have the potential to resolve completely [41,49,130], particularly with treatment [49] (Figure 8a,b). While linear opacities may clear, Murdoch and co-workers found an increased likelihood of progression over time and more so than with other morphologic features [41]. The natural history of ground-glass opacities is more difficult to predict and this CT pattern is a poor predictor of both disease activity and prognosis [41,49]. In part, this might be due to the non-specificity of CT ground-glass opacification which might indicate ‘active’ (and therefore potentially reversible) granulomatous inflammation or irreversible fine fibrosis below the limits of CT resolution [50]. CT abnormalities tending to indicate irreversible disease include reticulation, architectural distortion, honeycombing and traction bronchiectasis/bronchiolectasis. However, while some patients inevitably progress despite treatment, relative stability over time is more common in sarcoidosis-related ILD than in overtly progressive fibrotic DILDs [108].

While risk factors such as black race and female sex have been associated with higher rates of fibrotic pulmonary sarcoidosis [131,132], there are no formal, large-scale studies that have identified reliable morphological predictors on CT. This may relate to the high prevalence of asymptomatic disease [133] and the fact that patients are rarely observed to progress from one recognisable ‘stage’ to another. In the authors’ experience, fibrotic sarcoidosis often presents with imaging features that appear disproportionately severe when compared to symptoms and functional profiles.

### 7.3. Factors Contributing to and Predictors of Mortality in Sarcoidosis

Overall, the outlook for patients with pulmonary sarcoidosis is reasonably good with a mortality rate of 0.5–4.8% [134]. Lung fibrosis in sarcoidosis is a harbinger of ventilatory impairment leading to respiratory failure and death [109,135]. A study by Nardi et al., focussing on a subgroup of 142 patients with fibrotic pulmonary sarcoidosis, reported mortality as high as 11.3% with a mean age at death of just 55.2 years [110]. Pulmonary hypertension is an important independent predictor of mortality and, in the context of sarcoidosis, has a 5-year survival rate of only 55% [109,136,137]. The prevalence of sarcoidosis-associated pulmonary hypertension is higher in those with fibrosis but correlates poorly with the extent of abnormality on CT; moreover, nearly one-third of SAPH cases have no evidence of fibrosis on CT [95,100,136].

The utility of CT coupled with physiologic indices (including the composite physiological index (CPI) which was first developed in IPF [119]), has been explored as a ‘staging’ system to predict mortality in sarcoidosis [138]. In this system, a CPI threshold of 40 units was combined with the mean pulmonary artery to ascending aortic diameter ratio and an extent of fibrosis of more than 20% to form an algorithm which was significantly more predictive of outcome than any variable taken alone.

## 8. Summary

Imaging tests have an established place in the management of sarcoidosis. In patients with ‘classical’ appearances—either on CXR or CT—experienced radiologists will frequently offer a confident radiological diagnosis. In this regard, because of superior contrast resolution and the absence of anatomical superimposition, CT outperforms CXR. CT appearances in sarcoidosis vary considerably although expert opinion suggests that, among the apparently myriad different morphologic manifestations, there are recognisable CT phenotypes. Quantitative studies in which morphological abnormalities on CT are related to functional indices have provided unique insights into the pathophysiology of sarcoidosis and these have been discussed in the present review. Finally, the important role of CT in monitoring sarcoidosis has been presented.

## Figures and Tables

**Figure 1 jcm-13-00822-f001:**
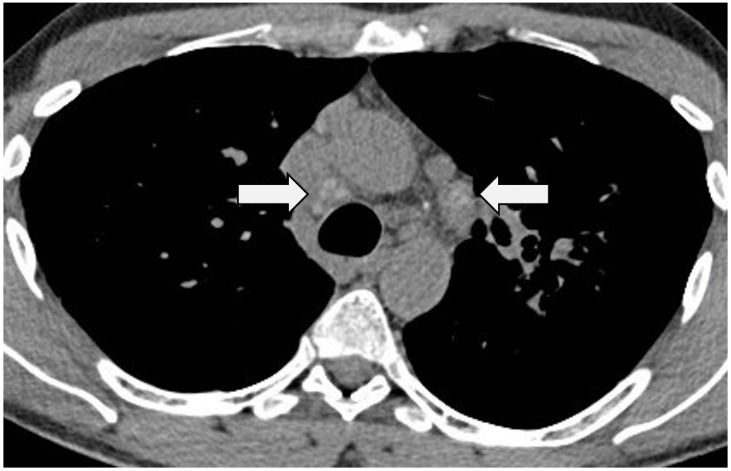
Axial CT in a patient with sarcoidosis. Images at a level below the aortic arch demonstrating classical ‘icing sugar’ calcification in mediastinal lymph nodes (arrows).

**Figure 2 jcm-13-00822-f002:**
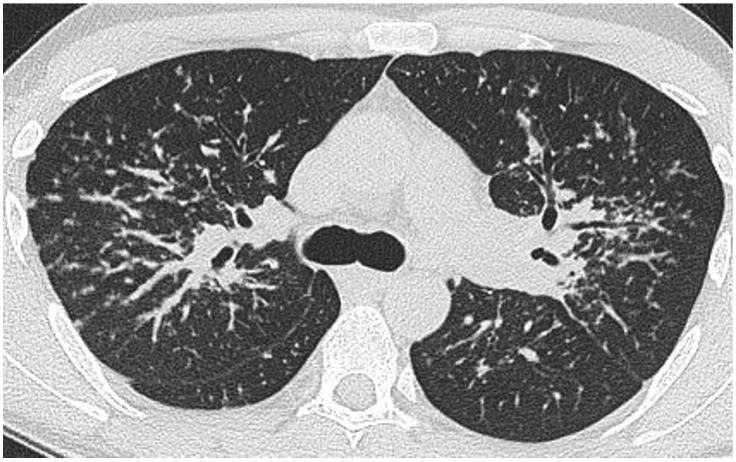
CT at the level of the carina in the same patient as in Figure 1. There is a roughly symmetrical bronchocentric micronodular infiltrate. More centrally, there is dense parenchymal opacification caused by conglomeration of nodules around the bronchovascular structures.

**Figure 3 jcm-13-00822-f003:**
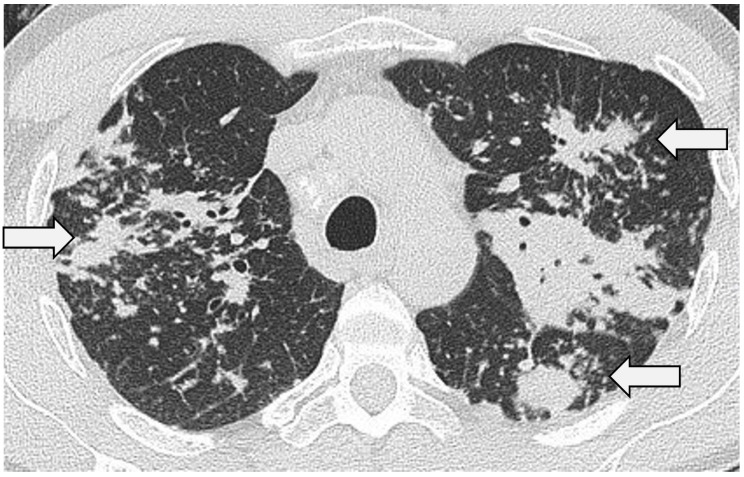
Nodular sarcoidosis in a 46-year-old male patient. CT at the level of the aortic arch showing large nodules with surrounding micronodules (the ‘galaxy sign’) in both upper lobes (arrows).

**Figure 4 jcm-13-00822-f004:**
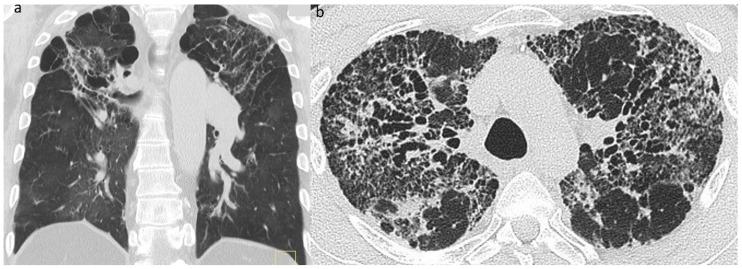
(**a**,**b**) Bilateral upper zone fibrosis with volume loss in two patients with sarcoidosis: (**a**) striking peri-bronchovascular fibrosis with retractile airway dilatation (i.e., traction bronchiectasis) and (**b**) CT through the upper lobes in a 64-yr-old male patient. Again, note the marked bronchocentric reticulation with severe traction bronchiectasis, which is particularly severe on the right.

**Figure 5 jcm-13-00822-f005:**
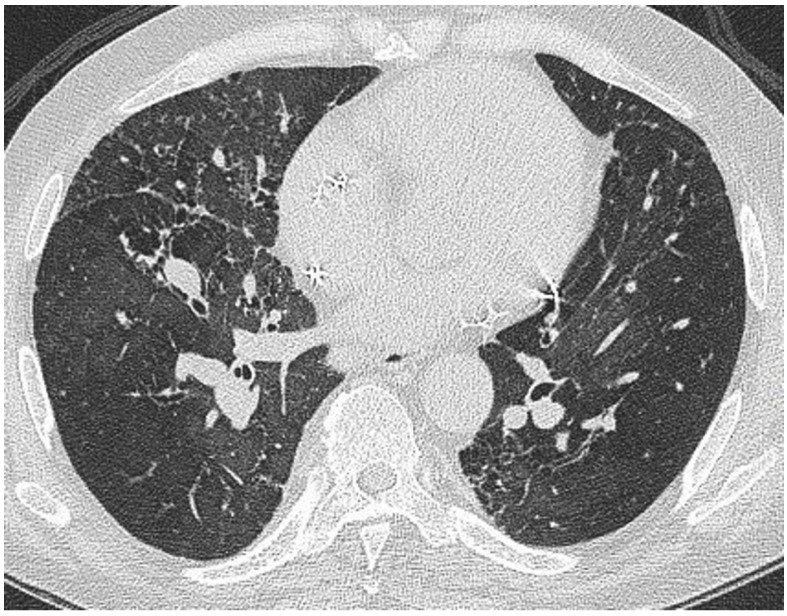
Small airway disease in sarcoidosis. Image through the lower zones shows a subtle but definite mosaic attenuation pattern; there is a reduction in the number/calibre of vessels within the lucent lung.

**Figure 6 jcm-13-00822-f006:**
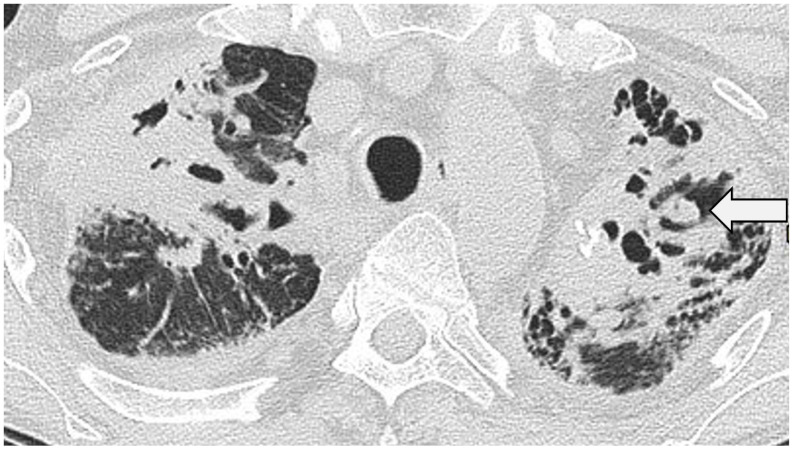
CT through the upper zones in a 61-year-old patient with pulmonary sarcoidosis. There is extensive disease with bronchocentric fibrosis manifest as a pattern of the consolidated lung. Note that in the left upper lobe, there is evidence of cavitation with a small aspergilloma (arrow).

**Figure 7 jcm-13-00822-f007:**
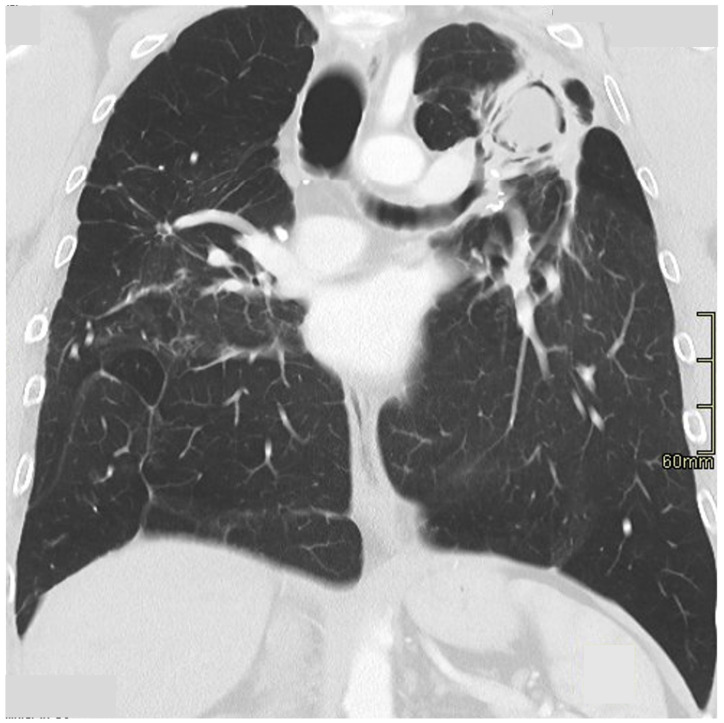
Coronal reconstruction of fibrocavitary disease in sarcoidosis; there is a large cavity in the left upper zone containing fungal material.

**Figure 8 jcm-13-00822-f008:**
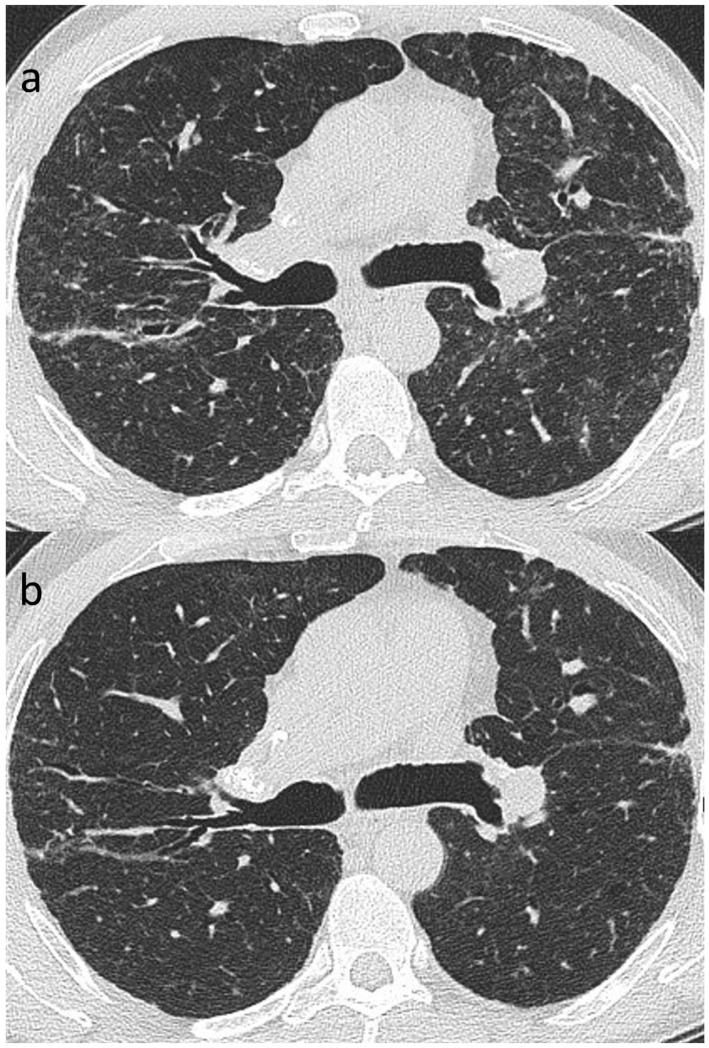
(**a**,**b**): Reversible disease in sarcoidosis. Targeted images of the left mid-zone showing the resolution of multiple random micronodules over time in (**a**) 2015 and (**b**) 2017.

**Table 1 jcm-13-00822-t001:** Principal reasons for CT monitoring in pulmonary sarcoidosis.

Principal Reasons for CT Monitoring in Pulmonary Sarcoidosis
To chart disease behaviour in patients with an initial ‘low confidence, provisional’ diagnosis of sarcoidosis in whom integration with serial PFTs and clinical features may modify diagnostic likelihoods.
To ascertain the likelihood of reversibility at baseline and/or during the natural course of the disease.
For the assessment of treatment response (including drug trials in sarcoidosis).
Prognostication based on the presence/absence of CT features (e.g., disease extent, traction bronchiectasis/bronchiolectasis and honeycombing).

**Table 2 jcm-13-00822-t002:** CT phenotypes in sarcoidosis based on the expert opinion of pulmonologists and thoracic radiologists [65].

CT Phenotype	Description
Non-fibrotic	Micronodular—peri-bronchovascular, peri-fissural and/or subpleural predilection, predominantly in the mid/upper zones, with or without a minority component of larger nodules with surrounding micronodules (i.e., ‘galaxy sign’), architectural distortion or volume loss
	Nodular (>3 mm but <3 cm)—peri-bronchovascular, peri-fissural and/or subpleural predilection, predominantly in the mid/upper zones, with or without a minority component of larger nodules with surrounding micronodules (i.e., ‘galaxy sign’), architectural distortion or volume loss
	Nodular (>3 mm but <3 cm)—random distribution
	Consolidation as the dominant or sole pattern
Likely to be fibrotic	Bronchocentric reticulation without cavitation and/or fibro-bullous destruction and with or without dense parenchymal opacification and/or a minority component of other CT abnormalities (e.g., delicate bands of ‘loose’ reticulation; enlarged peripheral pulmonary arteries, central pulmonary artery enlargement or a mosaic attenuation pattern)
	Bronchocentric reticulation with cavitation and/or fibro-bullous destruction and with or without dense parenchymal opacification and/or a minority component of other CT abnormalities (e.g., delicate bands of ‘loose’ reticulation; enlarged peripheral pulmonary arteries, central pulmonary artery enlargement or a mosaic attenuation pattern)
	Bronchocentric masses (‘progressive massive fibrosis [PMF]-lookalike’) with or without a minority component of other CT abnormalities (e.g., delicate bands of ‘loose’ reticulation; enlarged peripheral pulmonary arteries, central pulmonary artery enlargement or a mosaic attenuation pattern)

## Data Availability

No new data were created or analyzed in this study. Data sharing is not applicable to this article.

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
