# Peer review of "Imaging of Pulmonary Sarcoidosis—A Review"

_jcm, 2024, doi:10.3390/jcm13030822_

Round 1

Reviewer 1 Report

Comments and Suggestions for Authors

Review Journal of Clinical Medicine jcm-2832841 ‘Imaging in Pulmonary Sarcoidosis’ by Georgina Bailey, Sujal Desai and Athol Wells.

This review on imaging in pulmonary sarcoidosis is very well written and comprehensive. Furthermore, nice images have been used to support the text.  So it is a nice overview on this manifestation of sarcoidosis.

I do have some minor comments.

Minor comments

-      Regarding all the figures: sometimes it is mention in the legend that there are arrows to point out the abnormality, but the arrows are not visible (figure 1 and 3). And I would suggest adding arrows in figure 4.

-      Regarding figure 3: this is an image of bronchocentric fibrosis. I would suggest that this would be figure 7. Figure 7 is mentioned on page 8 in the section on pulmonary fibrosis. Figure 7 now is an image of fibrocavitary fibrosis and suggestion of an aspergilloma which is mentioned on page 9 as figure 9. However, there is no figure 9, so I think perhaps there is a mix-up of images (figure 3 should be figure 7 and figure 7 is figure 9). Only a new figure 3 should then be inserted.

-      Regarding figure 8; these are 2 images of reversible disease. In the text there is no reference to image 8. So perhaps on page 11 (part on disease monitoring) you can refer to image 8. It should be the last figure then in this review.

-      I am assuming that these images are all images of the authors’ own patient population. Do the authors have informed consent to use these images?

-      I would suggest to switch the order of table 1 and 2, since table 2 is now mentioned first in the text and table 1 thereafter.

-      One final comment on page 15 line 405-407. I would suggest to delete this part of the sentence (‘interestingly…the Japanese population’). Since the review is on pulmonary sarcoidosis and not cardiac sarcoidosis, to mention geographical differences in the Japanese population on cardiac sarcoidosis deaths is a bit out of context. The other factors do relate to pulmonary sarcoidosis.

Typos

-      Page 2 line 65: delete the first word ‘imaging’. Now it is mentioned twice in one sentence (‘the imaging first imaging tests requested..’)

-      Page 2 line 97-98: I would suggest to add ‘and’. So ‘tends to be bilateral and may have a focal pattern’.

-      Page 4 line 138: I would suggest to add ‘of’. So ‘a consequence of conglomerated granulomata’.  

Comments on the Quality of English Language

It is a very well written manuscript with only a few typo's (see my comments)

Author Response

Author’s reply to the Review Report: Reviewer 1

The author’s responses are written below the review comments in red italics.

-      Regarding all the figures: sometimes it is mention in the legend that there are arrows to point out the abnormality, but the arrows are not visible (figure 1 and 3). And I would suggest adding arrows in figure 4.

  • Arrows have been added to the figures mentioned above

-      Regarding figure 3: this is an image of bronchocentric fibrosis. I would suggest that this would be figure 7. Figure 7 is mentioned on page 8 in the section on pulmonary fibrosis. Figure 7 now is an image of fibrocavitary fibrosis and suggestion of an aspergilloma which is mentioned on page 9 as figure 9. However, there is no figure 9, so I think perhaps there is a mix-up of images (figure 3 should be figure 7 and figure 7 is figure 9). Only a new figure 3 should then be inserted.

  • The images have been re-ordered as suggested

-      Regarding figure 8; these are 2 images of reversible disease. In the text there is no reference to image 8. So perhaps on page 11 (part on disease monitoring) you can refer to image 8. It should be the last figure then in this review.

  • All images are now in the appropriate order and correctly referenced in the text

-      I am assuming that these images are all images of the authors’ own patient population. Do the authors have informed consent to use these images?

  • All images have been obtained from the patient population of the Royal Brompton Hospital, London. There is a policy at our institution that all anonymised images can be used for research purposes.

-      I would suggest to switch the order of table 1 and 2, since table 2 is now mentioned first in the text and table 1 thereafter.

  • The order of the tables has been changed accordingly

-      One final comment on page 15 line 405-407. I would suggest to delete this part of the sentence (‘interestingly…the Japanese population’). Since the review is on pulmonary sarcoidosis and not cardiac sarcoidosis, to mention geographical differences in the Japanese population on cardiac sarcoidosis deaths is a bit out of context. The other factors do relate to pulmonary sarcoidosis.

  • This part of the sentence has been removed

Typos

-      Page 2 line 65: delete the first word ‘imaging’. Now it is mentioned twice in one sentence (‘the imaging first imaging tests requested..’)

-      Page 2 line 97-98: I would suggest to add ‘and’. So ‘tends to be bilateral and may have a focal pattern’.

-      Page 4 line 138: I would suggest to add ‘of’. So ‘a consequence of conglomerated granulomata’.  

  • The above mentioned typographical errors have been corrected.

Reviewer 2 Report

Comments and Suggestions for Authors

Dear authors,

Thank you for submitting your manuscript titled “Imaging of Pulmonary Sarcoidosis – A Review” for review.

Despite the fact that sarcoidosis is a long-studied disease, the topic is relevant and consistent with the chosen title. The structure of the article is balanced and the information is presented in a fluent, easy-to-read manner. The only major inadvertence is related to the processing of the images, their matching in the text and, probably, the lack of some of them. I particularly appreciate the detailed presentation of CT aspects. The conclusions are objective and relevant from the point of view of clinical activity.

I have identified few areas where the manuscript could benefit from further enhancements. Below are my detailed suggestions:

-         Abstract and Keywords are missing

-         in order to respect the format of the journal, references must be registered between [ ]

-     Figure 1and Figure 3 – Arrows are mentioned in the legend but do not appear in the figures. Consider re-editing the images.

-          Figure 5 a,b - I suggest pasting the two images for better viewing

-          Lines 188-19: Figure 7 is not consistent with the text

-          Figure 8 - is missing from the text but appears as an image.

-          Lines 217-219: Figure 9 is missing

-         Line 269: ILD - “interstitial lung disease”. Abbreviation used for the first time without being detailed

-    Table 2 appears in the text before Table 1 – Line 312 and Line 339 respectively

-      All statements are missing: Author Contributions, Funding, Institutional Review Board Statement, Informed Consent Statement, Data Availability Statement, Acknowledgments, Conflicts of Interest

-          The format of the references does not correspond to the one accepted by the journal

I hope these suggestions will be helpful in strengthening your manuscript and better conveying the important research you have undertaken. Overall, my peer review is a major revision.

Looking forward to seeing the revised version of your work.

Best regards.

Author Response

Author’s reply to the Review Report: Reviewer 2

(The author’s responses are written below the review comments in red italics)

I have identified few areas where the manuscript could benefit from further enhancements. Below are my detailed suggestions:

-         Abstract and Keywords are missing

  • An abstract and keywords have been added

-         in order to respect the format of the journal, references must be registered between [ ]

  • The format of the references has been changed

-     Figure 1and Figure 3 – Arrows are mentioned in the legend but do not appear in the figures. Consider re-editing the images.

  • Arrows have been added to these images

-          Figure 5 a,b - I suggest pasting the two images for better viewing

  • The images have been pasted together for better viewing

-          Lines 188-19: Figure 7 is not consistent with the text

  • The images have been re-ordered and are now correctly referenced within the text

-          Figure 8 - is missing from the text but appears as an image.

  • The images have been re-ordered and are now correctly referenced within the text

-          Lines 217-219: Figure 9 is missing

  • The images have been re-ordered and are now correctly referenced within the text

-         Line 269: ILD - “interstitial lung disease”. Abbreviation used for the first time without being detailed

  • The detail has been added prior to the abbreviation

-    Table 2 appears in the text before Table 1 – Line 312 and Line 339 respectively

  • The order of the tables has been changed to appear correctly in the text

-      All statements are missing: Author Contributions, Funding, Institutional Review Board Statement, Informed Consent Statement, Data Availability Statement, Acknowledgments, Conflicts of Interest

  • The statements mentioned above have all been added

-          The format of the references does not correspond to the one accepted by the journal

The format of the references now corresponds to that accepted by the journal

Round 2

Reviewer 2 Report

Comments and Suggestions for Authors

Dear authors,

Thank you for re-submitting your manuscript titled “Imaging of Pulmonary Sarcoidosis – A Review”.  

After the changes made, I consider that the manuscript the quality of the manuscript improved in terms of both form and content. However, I detected that the position of Figures 5 and 6 is incorrect. Figure 5 shows an axial section at the level of the upper lobes with the cavity image and the possible aspergilloma. Figure 6 shows an axial section at the level of the lower lobes with a mosaic appearance.

In conclusion, I recommend minor revisions.

Author Response

The images have been swapped on the manuscript. Many thanks for your review.